# Characterization and Functionalization Approaches for the Study of Polymeric Nanoparticles: The State of the Art in Italian Research

Biagio Todaro [1] and Melissa Santi [2,*]

1   NEST Laboratory, Scuola Normale Superiore, Piazza San Silvestro 12, 56127 Pisa, Italy
2   NEST, Istituto Nanoscienze-CNR and Scuola Normale Superiore, Piazza San Silvestro 12, 56127 Pisa, Italy
*   Correspondence: melissa.santi@nano.cnr.it

**Abstract:** Polymeric nanoparticles (PNPs) are a group of nanocarriers employed in a wide range of applications. Characterization is a fundamental step in PNPs formulation and many basic techniques are available to provide chemical-physical information such as dimensions, surface potential, stability and solubility. Recently, these techniques have been implemented with more innovative ones to obtain more precise knowledge of the nanomaterials. In this review we analyzed the state of the art in the field of polymeric nanoparticles produced by Italian laboratories. We described all methods available for PNPs characterization with their applications as drug delivery systems. We also reported the different types of molecules that were recently used for PNPs functionalization, a fundamental step in delivering drugs specifically to their targets and then resulting in reduced side effects in patients.

**Keywords:** polymeric nanoparticles; drug delivery systems; characterization methods; targeting moieties; Italian research; Italy





## 1. Introduction

Many drugs used for the treatment of various human diseases can cause serious effects in patients when freely administered (e.g., chemotherapy drugs). The need to transport these molecules more efficiently to reduce side effects has led researchers to create nanoparticles-based drug delivery systems [1]. In general, nanoparticles present many advantages: (i) they improve the delivery of poorly water-soluble drugs; (ii) they reduce side effects; (iii) they can encapsulate more than one drug simultaneously; (iv) they increase drug stability; and (v) they can be engineered to cross biological barriers such as the blood brain barrier (BBB) [2]. Many types of nanoparticles are studied by researchers and, among these, the polymeric nanoparticles (PNPs) have gained increased consideration in the last decades with more than 50,000 publications in the world in the last 10 years [3]. This growing interest is mainly due to their favorable properties such as biocompatibility, biodegradability and easily tunable attractive bio-mimetic characteristics [4]. In particular, we found approximately 2000 publications just in Italy in the same period, underling the concrete interest of researchers and their effective potential [5]. Indeed, PNPs find applications in many fields, from drug delivery systems for treatment of human diseases [6,7], to antibacterial agents [8,9], water treatments [10] and cosmetics [11,12]. Polymers used for their synthesis are classified as synthetic or natural based on their origin, and many of them have already been approved by the Food and Drug Administration (FDA) [13]. The availability of numerous types of polymers and their combinations guarantee the possibility of obtaining nanoparticles with specific properties, such as controlled drug release and the consequent reduction of multiple administrations [14,15]. Despite these positive features, PNPs have not yet reached mass production for the treatment of diseases, mainly due to the lack of standardized protocols and the difficulty in translating them on a large scale [16]. In this review we described the state of the art in the formulation

of polymeric nanoparticles for the transport of drugs with a focus on studies carried out in Italy. We selected a subgroup of the most updated works of the last five years, starting from a sample of 2000 publications. To the best of our knowledge, this sample is representative for all the papers in this field of research which were carried out by researchers in Italian laboratories. In particular, we described the characterization methods applied in PNPs studies (Figure 1) and the strategies for the targeting of nanoparticles to specific sites of action. We reported all the techniques that have been used by research groups, from the most common to the most advanced, which can bridge the gap between pre-clinical and clinical research. Finally, due to the huge size of publications, we considered only those works that used exclusively polymeric nanoparticles avoiding composite nanomaterials like solid lipid nanoparticles or mixed metallic/polymeric nanoparticles.

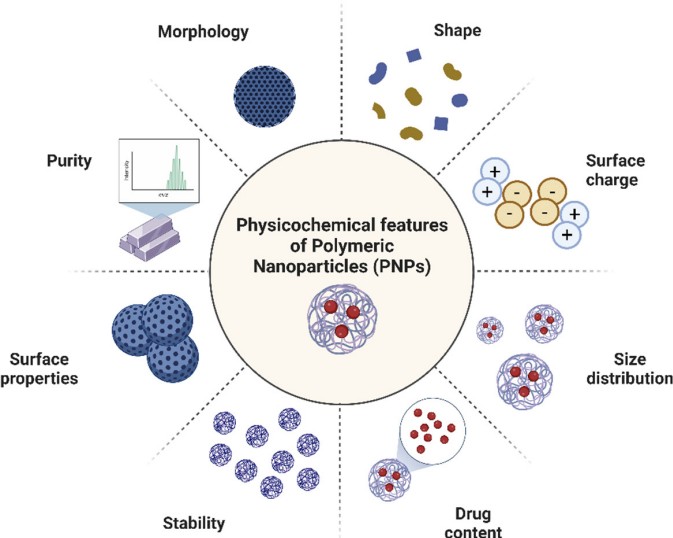

**Figure 1.** Graphical description of some of the pivotal physicochemical features of polymeric nanoparticles. The characterization process of nanoparticles requires a series of analyses which allow us to know the chemical composition, the surface characteristics, the dimension and the stability. All this information will be essential for the subsequent loading and transport of molecular payloads. Created with BioRender.com (accessed on 2 December 2022).

## 2. Methods Used for Nanoparticles Characterization

The analytical characterization of polymeric NPs has a pivotal role in terms of manufacturing process control, but also to establish issues concerning in vivo performances and toxicology [17]. PNPs properties, including size, electrostatic surface potential (termed Z potential or $\zeta$), Polydispersity Index (PDI), Drug Loading (DL), Encapsulation Efficiency (EE), stability, and drug release (DR) are usually assessed through several methods. The most common ones are mainly represented by Dynamic Light Scattering (DLS), Differential Scanning Calorimetry (DSC), Transmission Electron Microscopy (TEM), Scanning Electron Microscopy (SEM), Thermogravimetric Analysis (TGA), ultraviolet–visible (UV-vis) spectroscopy, Fourier Transform Infrared (FTIR), Circular Dichroism (CD), electrophoresis and chromatography [16]. In the quest for better sensing and thanks to the continuous technological development, Energy Dispersive X-ray (EDX), X-ray diffraction (XRD), X-ray Photoelectron Spectroscopy (XPS), Atomic Force Microscopy (AFM), nanoparticle tracking analysis (NTA), Single Particle Extinction and Scattering (SPES), Nuclear Magnetic Resonance (NMR), photoluminescence, Raman, and Fluorescence correlation spectroscopy (FCS) have great advantages, especially regarding the sensibility and the accuracy of the measures [17]. All of these methods for evaluating manufactured nanomaterials have been extensively and described in-depth in several reviews [18,19]. Trying to attempt all characterization techniques would be too huge and inaccurate work, hence this section will present only the characterization methods commonly used in Italian laboratories (Table 1).

**Table 1.** Principal techniques for the evaluation of the physicochemical features of polymeric nanoparticles.

| Characterization Techniques | NPs Characteristics Analyzed | References |
|---|---|---|
| Atomic Force Microscopy (AFM) | Size and size distribution<br>Shape of nanomaterials<br>Structure of nanomaterials<br>Surface charge of nanomaterials<br>Aggregation Surface properties<br>Dispersion of nanomaterials | [20–22] |
| Circular Dichroism (CD) | Structure of nanomaterials<br>Stability of nanomaterials<br>Surface properties of nanomaterials (coupled with ELISA)<br>Protein corona (conformation) | [23–25] |
| Dynamic Light Scattering (DLS) | Hydrodynamic size and size distribution<br>Protein corona (thickness and density) | [26–34] |
| Energy Dispersive X-Ray (EDX) | Composition of nanomaterials<br>Concentration of nanomaterials | [35] |
| Fluorescence Correlation Spectroscopy (FCS) | Critical association concentration determination<br>Size and size distribution<br>Drug content<br>In vitro drug release | [36–38] |
| Fourier Transform Infrared Spectroscopy (FTIR) | Structure and conformation of bioconjugates<br>Functional group analysis<br>Surface properties of nanomaterials | [21,22,39–44] |
| Nuclear Magnetic Resonance (NMR) | Structure of nanomaterials<br>Composition of nanomaterials<br>Purity of nanomaterials<br>Conformational change | [37,40] |
| Nanoparticle Tracking Analysis (NTA) | Size and size distribution | [20] |
| Raman Spectroscopy (RS) | Structure and conformation of bioconjugates<br>Functional group analysis | [22] |
| Scanning Electron Microscopy (SEM) | Size and size distribution<br>Shape of nanomaterials<br>Aggregation of nanomaterials | [39–42,45–48] |
| Single Particle Extinction and Scattering (SPES) | Size and size distribution | [49] |
| Thermogravimetric Analysis (TGA) | Drug-polymer interactions<br>Stability of nanomaterials<br>Purity of nanomaterials | [21,22,39,41,50] |
| Transmission Electron Microscopy (TEM) | Size and size distribution<br>Shape heterogeneity of nanomaterials<br>Dispersion/Aggregation of nanomaterials<br>Protein corona (thickness and density) | [21,23,36,41–43,47,50–52] |
| Ultraviolet–Visible Spectroscopy (UV-vis) | Size and size distribution | [23,24] |
| X-Ray Photoelectron Spectroscopy (XPS) | Surface properties of nanomaterials | [44] |
| X-Ray Diffraction (XRD) | Size and size distribution<br>Shape of nanomaterials<br>Structure of nanomaterials | [22,47,48] |
| Zeta Potential | Stability of nanomaterials<br>Surface charge of nanomaterials | [21,26] |

## 2.1. Particle Size Distribution and Morphological Characterizations

Particle size is the external dimension of a particle, thus it has a pivotal role both from the physico-chemical and pharmacological point of view. Particle size distribution can be used not only to monitor the stability of polymeric colloidal suspensions, but also to understand which processes are involved in the particles' transport, distribution and clearance in vitro and in vivo [53]. Size can be calculated both from physical properties, such as settling velocity, diffusion rate or coefficient and electrical mobility, but also exploiting microscope images using parameters such as the Feret diameter, the Martin diameter and

projected area diameter [54]. PNPs may generally have mean diameters between 100 and 300 nm, the polydispersity index ideally nearly zero in value, and present a unimodal size distribution. The nanoparticle size can be measured by using different techniques, but the most commonly used are the dynamic (DLS) and static (SLS) light scattering. DLS and SLS measure the oscillations in scattered light intensity caused by NPs moving under Brownian motion. Extensive literature on these techniques is available, especially because they are fast and nondestructive approaches that can give researchers a first idea about their formulations [26–34]. However, these procedures might furnish inaccurate measurements and slightly overestimate size in particular conditions: (i) in cases of aggregation and hydrophilicity, since the hydrodynamic-solvated state diameter is measured instead of the dry state diameter (such as in TEM); and (ii) in cases of protein corona formation, since serum, blood or other biological media could trigger significant modifications of the size of the suspended PNPs [55]. Thus, deeper investigation of PNPs' size may be performed by means of Nanoparticle Tracking Analysis (NTA) or Single Particle Extinction and Scattering (SPES). The former has brought significant progress in this field. NTA exploits the properties of both light scattering and Brownian motion in order to obtain particle size distributions of samples in liquid suspension. With a lower concentration detection limit compared to DLS, NTA may be very helpful in case of biological systems such as proteins, DNA and liquid media that relates the Brownian motion rate [20]. On the other side, SPES represent a valid optical alternative that allows a reliable distinction of the different species present in a solution, since each different population is analyzed separately, removing the protein corona influence [49].

Along with size distribution, the morphological characterization of PNPs has great interest since it provides many information about PNPs at a nanoscale level, especially revealing the shape and the presence of small cavities and pores. Although there are several different microscopic techniques, Scanning Electron Microscopy (SEM) [39–42,45–48] and Transmission Electron Microscopy (TEM) [21,23,36,41–43,47,50–52,56] are the main methods to perform such studies. The SEM technique is based on the electron scanning principle, which is useful to study the morphology and the dispersion of NPs in the bulk or matrix. Similarly, TEM is based on the electron transmittance principle, so it can furnish reliable and high-resolution images of PNPs at the nanometer level in the dry state, providing information on the morphology and shapes of the bulk nanoparticle from very low to higher magnifications. However, this technique requires sample preparation that may cause some artifacts, such as aggregation. Along with TEM and SEM, Atomic Force Microscopy (AFM) represents a valid alternative to characterize the surface morphology of PNPs. AFM allows for the measuring of a wide plethora of physical properties, such as magnetic fields, surface potential, surface temperature, friction, and many other features. Only three examples from Italian laboratories exploiting AFM are currently present in the literature, underling the innovation of this technique. In particular, AFM was used in the works from Villetti et al. and Rebanda et al. to measure maltoheptaose-b-polystyrene NPs size and copolymer Poly(L-lactide-co-caprolactone-co-glycolide) NPs size, respectively [20,21]. In a third work, the morphology of bio-nanocomposite-grafted paper sheets from Adel et al. [22] are determined by AFM. In addition, an ultraviolet–visible spectrophotometer can provide information about concentration, size, and shape by measuring the optical absorption, transmittance and reflectance. Ultraviolet–visible (UV–Vis) spectroscopy is principally exploited (i) to determine the doxorubicin release from hydrogels and nanogels by using calibration curves obtained by measuring absorbance at 480 nm and (ii) to demonstrate that the Enantiopure polythiophene nanoparticles display different antibacterial activities by measuring the optical density of *E. coli* and *S. aureus* suspensions at 600 nm [24,25].

### 2.2. Chemical Composition and Structural Characterizations

The chemical and structural characterizations are of primary importance to study the PNPs, providing several information on the bulk properties and on the composition and nature of bonding materials. Depending on the type of polymers, drugs and surfactants

components, and depending on the fingerprint region, which provides signature information about the material, different methods can be used to study the chemical and structural composition of polymeric NPs [17]. In this contest, the most common techniques are spectroscopic ones that are based on electromagnetic fields. They may differ according to the frequency at which these fields vary and how the signal is detected. The most developed and feasible techniques are X-ray spectroscopy, Fourier Transform Infrared (FTIR) spectroscopy, Raman spectroscopy, Circular Dichroism (CD) and Fluorescence correlation spectroscopy (FCS).

There is a significant body of literature on X-ray technologies, especially Energy Dispersive X-ray (EDX), X-ray diffraction (XRD) and X-ray Photoelectron Spectroscopy (XPS) [57]. EDX, which is normally fixed with a field emission SEM or TEM device, is widely used to determine the NPs elemental composition. The electron beam focused over a single NP permits the emission of characteristic energy X-rays, whose intensity is directly proportional to the concentration of the elements in the particle [58]. For instance, this technique was used by Gigli et al. to support SEM analysis for the confirmation of the presence of specific elements in a prepared amperometric biosensor based on Tyrosinase/Chitosan Nanoparticles [35]. XRD is used to analyze the crystallinity and phase of NPs [59]. The crystallinity index and the apparent crystallite sizes are usually determined from the Segal empirical equation and Debye-Scherrer equation, from which a rough idea about the particle size is also provided [60]. Nevertheless, in the case of PNPs having more amorphous characteristics with different /variable interatomic lengths, the acquisition and correct measurement may be difficult. XRD was mainly used to characterize polyethylene terephthalate nanoparticles from Lionetto et al., curcumin nanoparticles from Bilia et al., and to investigate the effects of alginate/oxidized nanocellulose-silver nanoparticles bio-nanocomposite on the structure and the morphology of the paper sheets from Adel and coworkers [22,47,48]. XPS is extensively used for the quantitative and qualitative characterization of the chemical composition of surfaces of a wide variety of materials. Since it is a non-destructive and very sensitive technique, it can be used in depth profiling studies to determine the overall composition of PNPs. Some of the XPS applications include the determination of the oxidation state of the catalysts and nanomaterials and a surface analysis of the organic coatings [16]. XPS analysis has been carried out only to gain information on the surface chemical composition of nanoparticles based on N,O-Carboxymethylchitosan-DA amide conjugate from Trapani et al [44].

Fourier transform Infrared (FTIR) and Raman spectroscopy are other vibrational spectroscopies useful to investigate the structural properties of PNPs and the molecular interactions between drugs and the encapsulating polymers. FTIR and Raman are complementary methods, as several strong vibrations permitted in the infrared region give weak Raman strips. With these techniques it is possible to confirm the presence of known compounds in the fingerprint region and detect the presence of impurities or unexpected interactions from the presence of characteristic functional groups [61]. A work from Adel et al. shows the synergy between these techniques on the comparison between paper sheets that are untreated and those that are coated by alginate/oxidized nanocellulose-silver nanoparticles [22].

Conversely, Fourier transform Infrared spectroscopy finds a wide range of applications, and some works are listed below. For instance, FTIR was used for (i) the identification of free components of Hyaluronic Acid-Decorated Chitosan Nanoparticles for the CD44-Targeted Delivery of Everolimus as well as their interaction during the NPs' formation [42]; (ii) the analysis and comparison of different Poly(l-lactide-co-caprolactone-co-glycolide)-Based nanoparticles loaded with doxorubicin or the hydrophobic SN-38 for cancer therapy [21]; (iii) the definition of the chemical structure of a hydrophilic antimicrobial peptide loaded on PEG-PLGA nanoparticles [43]; (iv) demonstrate the presence of graft poly($\varepsilon$-caprolactone (PLC) chains in the synthesis of cellulose-PLC copolymer [41]; (v) to observe the success of the paper sheets functionalization with the alginate/oxidized nanocellulose silver nanoparticles and to study changes in the structure and the morphology of these paper sheets [22]; (vi) to characterize the reaction intermediates and the final $\alpha$-Tocopherol–Chitosan NPs [40]; (vii) to verify the amorphous solid state features of N,O-Carboxymethylchitosan-DA amide

conjugate NPs for the nose-to-brain dopamine delivery [44]; and (viii) to confirm the effective nanoencapsulation of celecoxib or curcumin into the silk fibroin nanocarrier for osteoarthritis and to assure the conformational change of silk fibroin [39].

Circular dichroism (CD) spectroscopy is one of the main methods for the investigation of chiral compounds or achiral molecules assembled into complex structures of a chiral geometry [62]. It may also be used for the rapid evaluation of the folding, the binding and the conformational properties of a wide range of optically active molecules, from small molecules to natural (DNA, RNA and proteins) or synthetic macromolecules [62]. As for other types of nanoparticles, a work from Palamà et al. shows that enantiopure polythiophene PNPs in solution exhibited a markedly enhanced mirrored Cotton effect with respect to the polymers in solution for comparable absorption intensity and the absorption and CD spectra of the NPs, varied with the size of the nanoparticles themselves [25]. Other works exploit Far-UV CD spectra to evaluate the secondary structure of the released antitumoral trastuzumab from PLGA NPs or to assess the gelation kinetics of doxorubicin filled hydrogels and nanogels [23,24].

Fluorescence correlation spectroscopy (FCS) is considered another key technique for evaluation of some physicochemical characteristics of PNPs, such as the critical association concentration, the drug content and the in vitro drug release, exploiting the absorption or emission capacity of the materials [19]. Fluorescence allows the performance of the in-depth characterization of: (i) PEG-PLC di-block copolymers NPs surface [37]; (ii) to demonstrate that sucrose is a useful excipient during the docetaxel-loaded PLGA-PEG NPs preparation process and that it effectively cryoprotects nanoparticles [38]; and (iii) to measure the average diameter and polydispersity index of the blank-unlabeled and fluorescent-labelled chitosan-shelled/decafluoropentane-cored oxygen-loaded nanodroplets (OLN) formulations [36]. Furthermore, fluorescence may be coupled with 1H-NMR, especially for polymer functionalization characterization. For instance, this synergy is evident in two works from Trombino et al. and Venuta et al., where it allows the confirmation of the presence of $\alpha$-tocopherol on chitosan NPs or to evaluate the amount of PEG on the surface of poly($\varepsilon$-caprolactone) (PLC) NPs, respectively [37,40].

The purity of nanomaterials, the nature of polymer-drug interactions, as well as the polymer type and its physicochemical properties must be taken into account, especially to improve drug loading efficiency and control drug release. The amorphous or crystalline state of nanoparticle components and their interactions may control the solubility among these materials. Weak interactions between components generally cause drug crystallization because the drug and the polymer have a preference to interact with the molecules of the same type. Conversely, strong van der Waals forces and hydrogen bonds interactions between the drug and polymer allows the complete solubilization of the drug material in the polymer matrix, thus avoiding drug crystallization and achieving high drug loading efficiency [16]. Thermogravimetric Analysis (TGA) is routinely used to investigate the possible interactions between the NPs components (polymer, surfactants, drugs), usually by comparing the behavior of the loaded NP with the free components and the stability of the whole system. In TGA, the mass of the sample is monitored as a function of the heating suffered, and the result is a decomposition curve whose analysis gives the oxidation temperature and the residual mass of the sample [63]. Although the high costs and the destructive features limit the use of this technique, many works reported the use of TGA coupled with other techniques for different purposes: (i) to prove the presence of super-paramagnetic iron oxide nanoparticles inside poly(methyl methacrylate) nanoparticles and to quantify the difference between the nanohybrid particles [50]; (ii) with polymeric nanoparticles made of the copolymer Poly(L-lactide-co-caprolactone-co-glycolide), hosting two chemotherapeutic drugs [21]; (iii) to confirm the synthesis of sodium diclofenac loaded-self-associating cellulose-graft-poly($\varepsilon$-caprolactone) NPs [41]; (iv) to record simultaneously the thermal curves for empty and coated paper sheets decorated with alginate/oxidized nanocellulose-silver nanoparticles bio-nanocomposite [22]; (v) to evidence the stability of a chitosan solid nanoparticle formulation under non-oxidative conditions [44]; and (vi) to

record the mass loss due to drug decomposition of celecoxib or curcumin loaded- silk fibroin nanoparticles for osteoarthritis [39].

## 3. Targeting Strategies for PNPs Delivery

As a major condition for the design of NPs for drug-delivery purposes, the biological barriers for the in vivo distribution and the physical-chemistry nanoparticle profile need to be precisely analyzed. Once nanoparticles are injected into the bloodstream, they can be spread all over the body and only a minimal percentage finally reach their destination. In cancers, the presence of junctions (around 200 nm in diameter) inside the blood vessels and damaged endothelial structures, along with a lacking or dysfunctional lymphatic system and the inadequate drainage, facilitate passive accumulation of nanoparticles (enhanced permeability and retention effect, or EPR effect) [64]. However, EPR is not sufficient to avoid the spread of nanoparticles to other sites, mainly the liver and spleen, due to the continuous cleaning of the blood from foreign agents. Indeed, when nanoparticles are injected into the bloodstream they are immediately covered by serum proteins to form the so-called "protein corona" [65]. The protein corona drastically affects nanoparticle stability and targeting capabilities and activates the complement recognition by macrophages, which in turn causes the fast clearance of nanoparticles from the organism [65]. To overcome these problems, the functionalization of nanoparticles could promote their accumulation in specific sites and may reduce and modulate the formation of the protein corona, increasing the circulation time and PNPs stability [66]. Over the years, various approaches were used to decorate nanoparticles, and they fall mainly into four categories based on type of molecules used for the functionalization (Figure 2): (i) monoclonal antibodies (mAbs); (ii) oligonucleotide aptamers; (iii) peptides; and (iv) other molecules.

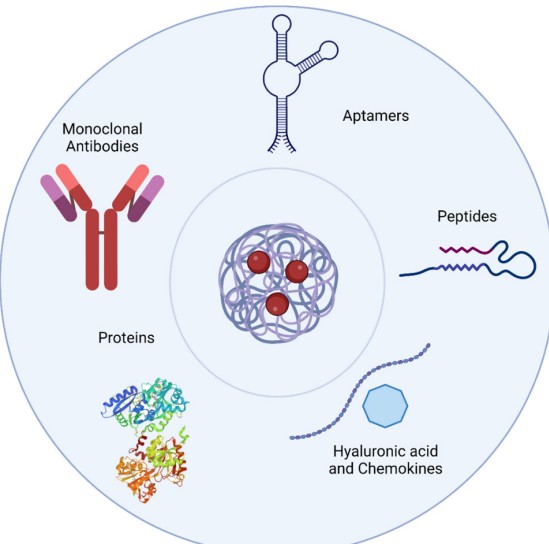

**Figure 2.** Types of targeting molecules available for PNPs functionalization. Different kinds of molecules can be used for the targeted delivery of nanoparticles. The main classes are monoclonal antibodies, oligonucleotide aptamers and other molecules such as like proteins or chemokines. Created with BioRender.com (accessed on 2 December 2022).

### 3.1. Monoclonal Antibodies

Monoclonal antibodies (mAbs) are molecules that present a great affinity for their targets and are widely used to functionalize nanoparticles. They are proteins formed by four chains defined as light and heavy chains based on their molecular weight, which is bound together by disulfide bridges. Light chains are responsible for target recognition, while heavy chains can be exploited for conjugation with nanoparticles [67]. Recently, Duskey, J.T. et al. demonstrated that PLGA nanoparticles labeled with the cell surface vimentin antibody (M08) showed high specificity for glioblastoma multiforme cancer cells

in comparison to normal ones [68]. Despite the greater affinities for their target with respect to other classes of molecules, the main limitations showed by monoclonal antibodies are their high production costs and their ability to cause an immune reaction [69].

### 3.2. Oligonucleotide Aptamers

Oligonucleotide aptamers are a class of molecules composed of DNA or RNA with unique conformations that allow them to specifically bind their targets [70]. Their affinities are usually lower than monoclonal antibodies, but they are cheaper and easily chemically modifiable in order to increase their stability and insert additional functional groups [71]. For these reasons they are widely used in nanoparticle functionalization. For instance, a group recently developed PNPs decorated with aptamers for the treatment of triple negative breast cancer (TNBC), a hard to treat form of breast cancer with a high risk of developing drug resistance, and with a high recurrence [72]. In these studies, two different approaches based on the delivery of cisplatin or siRNA to breast cancer cells by means of PLGA nanoparticles decorated with two different aptamers were developed, with the aptamers identified through the SELEX process [73,74].

### 3.3. Peptides

Peptides, like oligonucleotide aptamers, are short aminoacidic molecules with affinity for a specific target. They recently gained increasing attention due to their low costs of production, high yields and greater stability despite their typically lower targeting affinities than oligonucleotides and mAbs [75]. Their versatility and ease of conjugation led many research groups to employ them as targeting agents for the transport of polymeric nanoparticles [76]. Targeting peptides were applied for a variety of human diseases such as cancer, brain diseases and other disorders. For cancer therapy, many groups used RGD tripeptide for the specific delivery of PNPs to cancer cells [77–80]. RGD is widely used and is composed of a cyclic peptide of five amino acids with high affinity for $\alpha_5\beta_3$ integrin which is overexpressed in many tumors [78]. Other peptides are used for the targeted delivery of PNPs to the central nervous systems for the treatment of lysosomal storage disorders. For example in the work from Del Grosso et al., all tested peptides (Ang2, g7 and TF2) were able to cross the blood brain barrier (BBB) and allow nanoparticles to reach the brain for an improved enzyme replacement therapy both in vitro and in vivo (Figure 3) [81–83].

### 3.4. Other Molecules

Other molecules, such as entire or partial proteins can be conjugated to PNPs for targeted delivery of drugs. Recently, Boffi and co-workers developed a self-assembling ferritin-dendrimers nanoparticle for the treatment of myeloma [84]. Among all, chemokines are another class of molecules widely used as a targeting moiety to target cancer cells, employed by Pisani and co-workers to inhibit cell migration and metastasis formation [85]. Another very expended approach is the conjugation of nanoparticles with serum proteins such as transferrin (Tf). Tf is a serum protein responsible for the delivery of iron to cells through its receptor. Many solid tumors present the overexpression of the transferrin receptor (TfR) which in recent years has become one the most used targets for cancer therapy [86,87]. For instance, Tf-conjugated PNPs loaded with Docetaxel were developed by Souto and its collaborators for the treatment of metastatic breast cancer [88]. However, the conjugation of whole proteins at the nanoparticle surface does not allow for the precise control of the binding site on the protein itself, causing conformational changes in the protein which may decrease or lose the ability to recognize its target [89]. Finally, another interesting targeting molecule is hyaluronic acid (HA), which showed high affinity for CD44-overexpressing cells [90]. In particular Genta et. al, described the synthesis of HA-decorated chitosan nanoparticles loaded with everolimus for the treatment of bronchiolitis obliterans syndrome [42], while the work of Trevisan and co-authors showed the higher activity of HA-PNPs nanoparticles for the treatment of cancer cells [91].

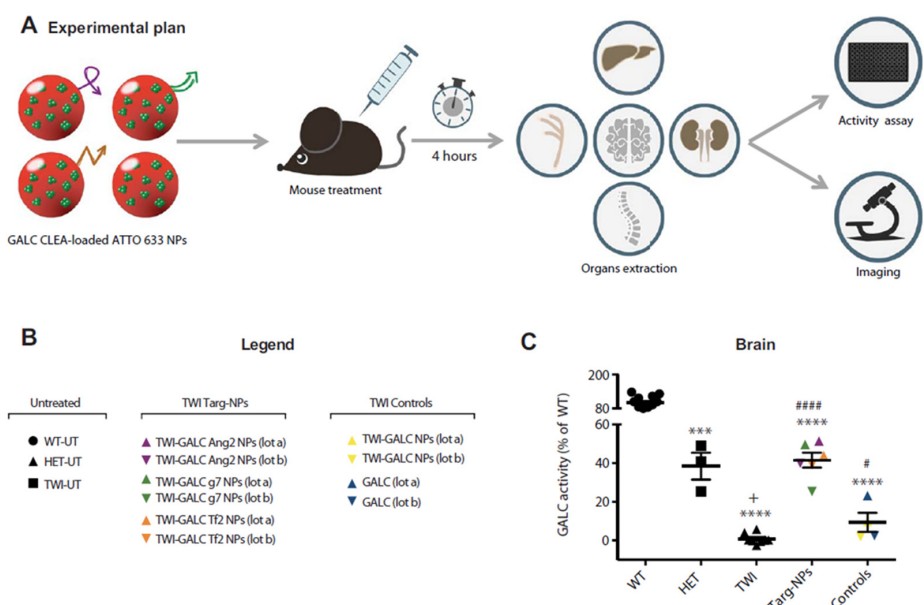

**Figure 3.** An example of targeting PNPs functionalized with different peptides for the targeting and crossing of the BBB. In this work, the authors showed the functionalization of PLGA nanoparticles with different peptides that are able to cross the BBB and performed an optimized enzyme replacement therapy in vivo. (**A**) Scheme of the experiment. Specific mice with Krabbe disease (TWI) were treated with targeted NPs (GALC Ang2 NPs, GALC g7 NPs, or GALC Tf2 NPs), nontargeted NPs (GALC CLEA NPs), or with the free enzyme rm-GALC (GALC). After four hours mice were euthanized, and GALC activity was assayed in extracted brain, sciatic nerves, spinal cord, kidneys, and liver by HMU-βGal assay. (**B**) Legend. (**C**) Brain GALC activity calculated for each animals. (From Del Grosso et al.) [81]. © The Authors, some rights reserved; exclusive licensee AAAS. Distributed under a Creative Commons Attribution NonCommercial License 4.0 (CC BY-NC) http://creativecommons.org/licenses/by-nc/4.0/ (accessed on 2 December 2022). *** $p < 0.001$ HET versus WT **** $p < 0.0001$ TWI, TWI targ-NPs, and TWI control versus WT # $p < 0.05$ TWI control versus TWI #### $p < 0.0001$ TWI-TARG-NPs versus TWI + $p < 0.05$ TWI versus HET.

## 4. Conclusions

This review describes the main targeting strategies for PNPs delivery and physicochemical properties of nanomaterials produced in Italy. Although the Italian research on this field has been extensive during the last years, a breakthrough of products in the market has not occurred so far. Indeed, on one hand, robust techniques for characterization along with biodegradability, stability and cytotoxicity are controlled by rigorous regulatory requirements for ensuring the safety of nanomedicines. On the other hand, the transition from lab-scale research to high-yielding production may be problematic, since the proposed polymeric nanoparticle must be cost-effective to produce. Moreover, although several approaches have been widely applied to produce polymeric nanoparticles, an available standard synthetic method is still necessary. Nanoprecipitation and methods based on emulsions are the most studied, although the usually broad size-distribution and the long exposure times to sonication, which forms radicals, are not always suitable for drug delivery applications. Moreover, the in vitro and in vivo kinetic release behavior, usually characterized by a rapid initial or burst release, need to be deeply understood before reaching clinical trials. Once these issues are overcome, nanomaterials will have the potential to impact physiological interactions from the nanoscale level to the systemic level, making nanomedicine an interesting research topic for the progress of current therapies.

**Author Contributions:** B.T. wrote the characterization section; M.S. wrote the functionalization paragraph; B.T. and M.S. contributed equally to the introduction and conclusions and revised the manuscript. All authors have read and agreed to the published version of the manuscript.

**Funding:** This research received no external funding.

**Institutional Review Board Statement:** Not applicable.

**Informed Consent Statement:** Not applicable.

**Data Availability Statement:** Not applicable.

**Acknowledgments:** We would like to thank Valerio Voliani who gave us the possibilities to use Bioren-der software to create some of the figures of this work and Stefano Luin for the fruitful discussions.

**Conflicts of Interest:** The authors declare that they have no conflicts of interest.

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
