# Peer review of "Characterization and Functionalization Approaches for the Study of Polymeric Nanoparticles: The State of the Art in Italian Research"

_2673-8023, doi:10.3390/micro3010002_

Round 1

Reviewer 1 Report

Biagio Todaro has reported paper on title synthesis of polymeric nanoparticles. the topic is very good related to nanotechnology. but there are many things to change it.

1. There is need to optimization of nanoparticles with respect to different condition such as temperature, pH, concentration of reactant and many more.

2. There is missing of Figure related to polymeric nanoparticles which can be more attractive for readers. 

3. There is need to discuss more factor inflecning on synthesis and characterization with new references.

  • DOI: 
  • 10.1021/acsomega.9b00119
  • DOI: 
  • 10.1166/jnn.2015.11908
  • DOI: 
  • 10.1007/s11051-014-2347-9

Author Response

Reviewer 1:

Biagio Todaro has reported paper on title synthesis of polymeric nanoparticles. the topic is very good related to nanotechnology. but there are many things to change it.

Answer: We thank the Reviewer for the valuable comments, and we guess her/his concerns are addressed in the following point-by-point response.

  1. There is need to optimization of nanoparticles with respect to different condition such as temperature, pH, concentration of reactant and many more.

Answer: Despite the optimization of temperature, pH and concentration of reagents are fundamental steps in the synthesis of nanoparticles, we would like to underline that this review is not focused on synthesis but only on the characterization and functionalization methods for polymeric nanoparticles. Moreover, this review is comprising in a specific issue which describe the work of Italian researches.

  1. There is missing of Figure related to polymeric nanoparticles which can be more attractive for readers. 

Answer: We thank the reviewer for his/her suggestion.

Action: We added another figure in the manuscript named Figure 1 which described the methods used for polymeric nanoparticles characterization.

  1. There is need to discuss more factor inflecning on synthesis and characterization with new references.

DOI: 10.1021/acsomega.9b00119

DOI: 10.1166/jnn.2015.11908

DOI: 10.1007/s11051-014-2347-9

Answer: We thank the reviewer for his/her suggestion. As mentioned before our manuscript is focused on characterization methods and not on synthesis procedures. We are sorry if our scope isn’t clear enough. Finally, suggested references describe lipid or solid lipid nanoparticles, which are not comprised in the scope of this review.

Action: We revised and improved the characterization paragraph and we modified the title of the manuscript to better underline that this work is focused on characterization and functionalization methods.

Reviewer 2 Report

This is a review intended for a scientific journal. However, it is written as a report for CNR or NEST. I can therefore recommend it for publication provided some changes have been made.

The topic itself is modern, well exposed and the review has a significant tutorial value.  To make it suitable for a scientific journal the authors should omit constructions like "Italian research", "Italian laboratories" etc.  They should only explain in one sentence that this is a review of the state of the art in the field of polymeric nanoparticles based on the sample of around 2000 publications produced by Italian laboratories. The justification for the sample should be any the authors might find reasonable. It might easily be that that sample is representative for all of the papers in this field of research (50000 of them in the same period).

Author Response

Reviewer 2

This is a review intended for a scientific journal. However, it is written as a report for CNR or NEST. I can therefore recommend it for publication provided some changes have been made.

Answer: We thank the Reviewer for the valuable comments, and we guess her/his concerns are addressed in the following point-by-point response.

1- The topic itself is modern, well exposed and the review has a significant tutorial value.  To make it suitable for a scientific journal the authors should omit constructions like "Italian research", "Italian laboratories" etc.  They should only explain in one sentence that this is a review of the state of the art in the field of polymeric nanoparticles based on the sample of around 2000 publications produced by Italian laboratories. The justification for the sample should be any the authors might find reasonable. It might easily be that that sample is representative for all of the papers in this field of research (50000 of them in the same period).

Answer: We thank the reviewer for his/her suggestion. We underline in the abstract and introduction the fact that the review is related to Italian researches. We also emphasized that in the review we sample a subset of papers that are included in the 2000 cited as examples in the introduction. In particular, we have restricted the research to the last 5 years trying to summarize only the most innovative researches due to the size of the sample. We have also limited the description to those works that used only 100% polymeric nanoparticles, eliminating hybrid nanomaterials such as solid lipid nanoparticles or metal nanoparticles conjugated with polymers.

Reviewer 3 Report

The authors presented the paper "Characterization and functionalization approaches for the synthesis of polymeric nanoparticles: the state of the art in Italian research."

1) Much more fresh 2-3 years paper should be presented. There are many 2022 year papers. The reference may be improved. 

2) Introduction section is poor. It doesn't show any problems in the area and don't highlight the content. The same concern is with Abstract too. I recommend inserting some more keywords.

3) I don't understand how is this review is related to Italy. Where the information about Italy starts?  Please mention it in Abstract, Introduction, may be some new section about the most important Italian researchers' contribution in the field. You present general things which are used not only in Italy.

4) The method section is not showing the methods features, for what these methods may be used. The information in Table 1 is too general, other methods may show the same thing, but what are the differences? Maybe, it will be better presenting some high qualitative pictures. For example, you may obtain the size by DLS, TEM, SEM, AFM, etc. DLS indicates the size with water shell. There are three size modes which are the number, volume, intensity.

5) The title of the review is "Characterization and functionalization approaches for the synthesis of polymeric nanoparticles". However, where is synthesis application? I see the characterization methods mostly. Some surface functionalization with address molecules. Vitamins? 

6) I recommend presenting some smart multimodal construction examples. Address molecules as vitamins and protein coating. But it is for your consideration.

7) Usually not only targeting of nanoparticles highlights. Potentially, some approaches of drug loading, reporter group surface functionalization? But it is for your consideration.

Minor comments

Table 1. It will be better to present it on one page.

Fig. 1 aptamers instead of oligonucleotides?

Author Response

Reviewer 3

The authors presented the paper "Characterization and functionalization approaches for the synthesis of polymeric nanoparticles: the state of the art in Italian research."

Answer: We thank the Reviewer for the valuable comments, and we guess her/his concerns are addressed in the following point-by-point response.

1) Much more fresh 2-3 years paper should be presented. There are many 2022 year papers. The reference may be improved. 

Answer: We thank the reviewer for his/her suggestion.

Action:  We carefully revised all references and add some new ones (reference number 9,12 and 16).

2) Introduction section is poor. It doesn't show any problems in the area and don't highlight the content. The same concern is with Abstract too. I recommend inserting some more keywords.

Answer: We thank the reviewer for his/her suggestions.

Action: We enriched the introduction and the abstract by adding a detailed description of problems related to the study of polymeric nanoparticles. Moreover, we added a new figure to better clarify which physicochemical features are analyzed in the manuscript (Figure 1).

3) I don't understand how is this review is related to Italy. Where the information about Italy starts?  Please mention it in Abstract, Introduction, may be some new section about the most important Italian researchers' contribution in the field. You present general things which are used not only in Italy.

Answer: We are sorry if this point is not clear enough. The main purpose of this work is to described the latest advances in Italian researches regarding characterization and functionalization methods of polymeric nanoparticles. Such methods are noticeably employed all over the world. We studied the literature and tried to report all the works conducted in Italy in the last five years and the majority of references are related to Italian researches and researchers. In our opinion a new paragraph about the most important Italian researchers’ contributions it may be too long to contain all the work involving polymer nanoparticles. Moreover, all works are already reported in the description of characterization and functionalization methods.

Action: We better highlight in the abstract and introduction that these review described only Italian works.

4) The method section is not showing the methods features, for what these methods may be used. The information in Table 1 is too general, other methods may show the same thing, but what are the differences? Maybe, it will be better presenting some high qualitative pictures. For example, you may obtain the size by DLS, TEM, SEM, AFM, etc. DLS indicates the size with water shell. There are three size modes which are the number, volume, intensity.

Answer: We thank the Reviewer for the valuable comment. The individual methods’ description is already reported in several other reviews. The table 1 is designed to give a general overview of the characterization methods used in Italy; we believe a more in-depth analysis does not reflect the guidelines of this work.

Action: We revised and improved the first section of method paragraph and we added two review P.C. Lin et coworker (reference 17) and A. Barhoum et al. ( reference16).

5) The title of the review is "Characterization and functionalization approaches for the synthesis of polymeric nanoparticles". However, where is synthesis application? I see the characterization methods mostly. Some surface functionalization with address molecules. Vitamins?

 Answer: We thank the reviewer to raise these points and we are sorry if the title was confusing. We decided to described only the characterization methods and functionalization of polymeric nanoparticles and not the synthesis to avoid a too long and unreadable review. Regarding vitamins functionalization, we carefully search in the available literature but we did not found matches on polymeric nanoparticles functionalized with vitamins or their derivatives. Vitamines are usually transported within NPs as cargo, since an “external delivery” may cause their degradation.

Action: We change the title to clarify the aim of the review and we added references.

6) I recommend presenting some smart multimodal construction examples. Address molecules as vitamins and protein coating. But it is for your consideration.

Answer: We thank the reviewer for his/her suggestion. We carefully check bibliography and we did not found any other papers about smart multimodal materials in the last five years reported by Italian researchers.

7) Usually not only targeting of nanoparticles highlights. Potentially, some approaches of drug loading, reporter group surface functionalization? But it is for your consideration.

Answer: As far as we understand, the reviewer asks to add a paragraph on the potential of nanoparticles in providing a useful tool for drug delivery, but in our opinion this topic is already addressed in the introduction. Regarding drug loading approaches, we believe that this point goes beyond the scope of this review.

Minor comments

Table 1. It will be better to present it on one page.

Answer: We thank the reviewer for his/her suggestion.

Action: We reduced the size of Table 1 as much as possible.

Fig. 1 aptamers instead of oligonucleotides?

Answer: We thank the reviewer for his/her suggestion.

Action: We change the word in the figure (now Figure 2)

Round 2

Reviewer 2 Report

The review has been improved in many aspects, but the the style has been left essentially unchanged. The authors did not respond correctly to my main objection which was also raised by Reviewer 3.

They even made the whole situation worse by writing "In this review we analyzed the state of the art in Italian research on PNPs focusing our analysis on works carried out by Italian researchers only."  meaning that, for example, as a guest researcher in an Italian laboratory I would not be qualified for this review and my work would be excluded.

I can here only repeat a part of my review "They should only explain in one sentence that this is a review of the state of the art in the field of polymeric nanoparticles based on the sample of around 2000 publications produced by Italian laboratories. The justification for the sample should be any the authors might find reasonable. It might easily be that that sample is representative for all of the papers in this field of research".

Author Response

Reviewer 2

The review has been improved in many aspects, but the style has been left essentially unchanged. The authors did not respond correctly to my main objection which was also raised by Reviewer 3.

They even made the whole situation worse by writing "In this review we analyzed the state of the art in Italian research on PNPs focusing our analysis on works carried out by Italian researchers only."  meaning that, for example, as a guest researcher in an Italian laboratory I would not be qualified for this review and my work would be excluded.

I can here only repeat a part of my review "They should only explain in one sentence that this is a review of the state of the art in the field of polymeric nanoparticles based on the sample of around 2000 publications produced by Italian laboratories. The justification for the sample should be any the authors might find reasonable. It might easily be that that sample is representative for all of the papers in this field of research".

Answer: We are really sorry if we didn’t understand the reviewer request. He/she is right and we agree with him/her, since the phrase “In this review we analyzed the state of the art in Italian research on PNPs focusing our analysis on works carried out by Italian researchers only."  is misleading. We would like to take this chance to better explain which was our process of selection and choice of the subject and works cited in the manuscript. Starting from the specific issue of the journal “State-of-the-Art Microscale and Nanoscale Researches in Italy” we did a first research finding 2000 papers on PNPs in the last 10 years as also reported in the introduction. From this big sample we developed our manuscript taking in consideration only the most representative works in the last 5 years to be also as updated as possible.

Action: We put our effort in order to satisfy the reviewer request by adding new sentences in the Introduction, stating that the sample is representative for all the papers in this field of research done in Italian laboratories by researchers (Italian and not).

Reviewer 3 Report

Thank you for the revised paper.

Author Response

Reviewer 3

Thank you for the revised paper.

Answer: We thank the reviewer for appreciating our improvements

Round 3

Reviewer 2 Report

To my opinion the paper is now valuable of reading to everyone interested in the subject of polymeric nanoparticles.